# The Specific Capsule Depolymerase of Phage PMK34 Sensitizes *Acinetobacter baumannii* to Serum Killing

**DOI:** 10.3390/antibiotics11050677

**Published:** 2022-05-17

**Authors:** Karim Abdelkader, Diana Gutiérrez, Agnieszka Latka, Dimitri Boeckaerts, Zuzanna Drulis-Kawa, Bjorn Criel, Hans Gerstmans, Amal Safaan, Ahmed S. Khairalla, Yasser Gaber, Tarek Dishisha, Yves Briers

**Affiliations:** 1Department of Biotechnology, Ghent University, Valentin Vaerwyckweg 1, 9000 Gent, Belgium; kareem.sofy@pharm.bsu.edu.eg (K.A.); diana.gutierrezfernandez@ugent.be (D.G.); agnieszka.latka@ugent.be (A.L.); dimitri.boeckaerts@ugent.be (D.B.); bjorn.criel@ugent.be (B.C.); hans.gerstmans@kuleuven.be (H.G.); 2Department of Microbiology and Immunology, Faculty of Pharmacy, Beni-Suef University, Beni-Suef 62511, Egypt; ahmedkhairalla@pharm.bsu.edu.eg (A.S.K.); yasser.gaber@pharm.bsu.edu.eg (Y.G.); tarek.dishisha@pharm.bsu.edu.eg (T.D.); 3Department of Pathogen Biology and Immunology, Institute of Genetics and Microbiology, University of Wroclaw, Przybyszewskiego 63, 51-148 Wrocław, Poland; zuzanna.drulis-kawa@uwr.edu.pl; 4Department of Data Analysis and Mathematical Modelling, Ghent University, Coupure Links 653, 9000 Gent, Belgium; 5Laboratory of Gene Technology, Department of Biosystems, KU Leuven, Kasteelpark Arenberg 21, 3001 Leuven, Belgium; 6Department of Biosystems, KU Leuven, Willem de Croylaan 42, 3001 Leuven, Belgium; 7Department of Microbiology and Immunology, Faculty of Pharmacy, Menoufia University, Shebin El-Koum 51132, Egypt; amaleisa_sb@yahoo.com; 8Department of Biology, University of Regina, Regina, SK S4S 0A2, Canada; 9Department of Pharmaceutics and Pharmaceutical Technology, Faculty of Pharmacy, Mutah University, Karak 61710, Jordan

**Keywords:** capsule, depolymerase, phage, *Acinetobacter baumannii*, tailspike, antivirulence

## Abstract

The rising antimicrobial resistance is particularly alarming for *Acinetobacter baumannii*, calling for the discovery and evaluation of alternatives to treat *A. baumannii* infections. Some bacteriophages produce a structural protein that depolymerizes capsular exopolysaccharide. Such purified depolymerases are considered as novel antivirulence compounds. We identified and characterized a depolymerase (DpoMK34) from Acinetobacter phage vB_AbaP_PMK34 active against the clinical isolate *A. baumannii* MK34. In silico analysis reveals a modular protein displaying a conserved N-terminal domain for anchoring to the phage tail, and variable central and C-terminal domains for enzymatic activity and specificity. AlphaFold-Multimer predicts a trimeric protein adopting an elongated structure due to a long α-helix, an enzymatic β-helix domain and a hypervariable 4 amino acid hotspot in the most ultimate loop of the C-terminal domain. In contrast to the tail fiber of phage T3, this hypervariable hotspot appears unrelated with the primary receptor. The functional characterization of DpoMK34 revealed a mesophilic enzyme active up to 50 °C across a wide pH range (4 to 11) and specific for the capsule of *A. baumannii* MK34. Enzymatic degradation of the *A. baumannii* MK34 capsule causes a significant drop in phage adsorption from 95% to 9% after 5 min. Although lacking intrinsic antibacterial activity, DpoMK34 renders *A. baumannii* MK34 fully susceptible to serum killing in a serum concentration dependent manner. Unlike phage PMK34, DpoMK34 does not easily select for resistant mutants either against PMK34 or itself. In sum, DpoMK34 is a potential antivirulence compound that can be included in a depolymerase cocktail to control difficult to treat *A. baumannii* infections.

## 1. Introduction

*Acinetobacter baumannii* is a notorious opportunistic pathogen responsible for up to 10% of the nosocomial infections caused by Gram-negative pathogens, including wound, blood, and urinary tract infections, and ventilator-associated pneumonia [1]. These infections generally have a poor prognosis, high morbidity, and high mortality [1]. Their treatment is hampered by the emergence and increasing spread of multi- and pandrug-resistant strains, leaving a limited number of therapeutic options available [2]. The overall fitness and virulence of this species is determined by the presence of a high molecular weight capsular polysaccharide (CPS) that surrounds the outer membrane (OM) [3]. CPS plays a key role in the pathogenicity of *A. baumannii* by acting as a protective barrier against harsh environmental conditions such as desiccation, assisting in the evasion of the host immune system, increasing resistance to antimicrobial compounds (antibiotics and disinfectants) [3] and being a major factor for exopolysaccharide (extracellularly secreted sugars contributing in biofilm formation) and biofilm formation [4], the latter contributing to recalcitrant and persistent infections. The *A. baumannii* capsule composition is highly diverse, determined by the specific capsule loci with more than 125 unique variants described for *A. baumannii* strains [5]. Previous studies have shown that CPS removal sensitizes *A. baumannii* to components of the immune system and reduces bacterial virulence in murine infection models [4,6,7], whereas upregulation of CPS production increases serum resistance and virulence [8].

In the search for alternative or complementary therapeutics, bacteriophage-encoded enzymes targeting the CPS have received increasing attention [5,9]. Many phages are equipped with tail-associated depolymerases that degrade CPS. The specificity of its capsular depolymerase is a primary determinant for the host spectrum of the phage [10]. Consistently, most characterized recombinant depolymerases have an overlapping spectrum with the phages they are derived from [11,12,13,14]. Nevertheless, a few studies report depolymerases that are also active on strains that are non-sensitive to the phage [15,16]. The antivirulence activity of capsule depolymerases, based on capsule removal, has been demonstrated both in vitro and in vivo [11,12,14,15,16,17,18,19,20,21,22]. Furthermore, in vivo application of capsule depolymerase sensitized *Klebsiella pneumoniae* strains to the action of standard-of-care antibiotics [23]. Purified depolymerase has also enhanced antibiofilm activity of polymyxin against *K. pneumoniae* [24].

In our previous work, we have isolated and characterized a lytic phage (vB_AbaP_PMK34) that specifically infects the extensively drug-resistant strain *A. baumannii* MK34 [25]. The plaques of phage PMK34 are surrounded by growing halos, a hallmark feature indicating depolymerase activity. This observation was further supported by the detection of a gene encoding a tailspike (*orf45*) with a pectate lyase conserved domain, which can act as a depolymerase [25]. In this study, we carried out an in-depth bioinformatic and functional characterization of this putative depolymerase, referred to as DpoMK34. Moreover, we assessed the antibacterial and antivirulence potential of DpoMK34. In addition, we evaluated the probability of resistance development against DpoMK34 upon repeated exposure.

## 2. Results

### 2.1. Identification and In Silico Characterization of DpoMK34

In our previous work, Acinetobacter phage PMK34 (vB_AbaP_PMK34) was isolated from the sewage water collecting system of the hospital of Beni-Suef (Egypt) [25]. Phage PMK34 produces clear lysis plaques surrounded by growing turbid halos upon prolonged incubation, suggesting the presence of phage-encoded depolymerase activity (Figure 1A,B). In the current study, negative staining of *A. baumannii* MK34 cells within the halo zone showed the removal of the capsule layer and disaggregation of the cell clumps, demonstrating that the halo zone results from capsule rather than lipopolysaccharide removal (Figure 1C,D).

Analysis of the open reading frames of the phage PMK34 genome (MN433707.1) resulted in the identification of a putative tailspike (*orf45*; QGF20174.1) encoding a putative depolymerase (DpoMK34) with a predicted molecular weight of 76 kDa and isoelectric point of 5.2. [25]. Multiple protein sequence alignment and domain analysis (Figure 2) revealed a three-domain architecture, which is featured by several other tail fibers and tailspikes from diverse phages [26]. The N-terminus of DpoMK34 (amino acids 15 to 110) is conserved among other tailspikes of *A. baumannii* phages and shows low similarity to the well-characterized N-terminus of the phage T7 tail fiber protein (PF03906; bit score 13.2; E-value 0.053). This so-called anchor domain is responsible for attachment of the T7 tail fiber to the phage tail, more specifically to the interphase of the adaptor (gp11) and nozzle protein (gp12) of the short tail complex [27]. Since phage PMK34 also belongs to the *Autographiviridae* family (T7 supergroup) [25], a similar role may be anticipated for the N-terminus of DpoMK34. The central domain of DpoMK34 (from amino acid 225 to 433) is predicted to belong to the pectate lyase_3 superfamily (PF12708; bit score 33.0; E-value 5.1 × 10^−8^, Figure 2). This domain is closely related to glycosyl hydrolase family 28 and has reported activities on CPS [11,12,14,15,18,28]. The C-terminus (from amino acid 434 to 699) does not show homology to a conserved domain (Figure 2). Other tailspikes or tail fibers with a three-domain architecture most commonly encode a C-terminal domain involved in host recognition or a chaperone domain for folding and trimerization. Chaperone domains are often autoproteolytically cleaved off [29,30,31,32,33].

The complete trimeric structure of DpoMK34 predicted by AlphaFold-Multimer [34], provides a more detailed insight in its modular composition (Figure 3). The elongated trimeric protein has three identical monomers with each monomer displaying a small N-terminal anchoring domain separated by a long α-helix from the central and C-terminal domain. The central domain comprises right-handed β-sheets orthogonal to the long axis that are packed together to create a β-helical domain, which is a typical topology for depolymerases [9] and is also observed in the crystal structure of the N-terminally truncated φAB6 tailspike protein (identity of 95.24%, query coverage of 100%; PDB accession number 5js4.1). The nomenclature of phage depolymerases as either tailspike or tail fiber (referring to the shape of the protein) is somewhat ambiguous. DpoMK34 appears to have hybrid properties with the typical elongated, fibrous structure similar to the T7 tail fiber (due to the long α-helix albeit shorter than in T7), and the enzymatic activity predicted in the central domain typical for tailspikes. For reasons of simplicity, we classify DpoMK34 here as a tailspike.

Many structural studies of tail fibers and tailspikes are often limited to the N-terminal or C-terminal fragments due to the difficulty to obtain crystals of the full-length protein (for example, the highly similar phiAB6 tailspike protein) [35]. AlphaFold-Multimer [34] thus offers new opportunities to explore the full, predicted modular structure. Nevertheless, access to supercomputing resources was needed to obtain predictions for this long and trimeric protein. While AlphaFold excels in accurate predictions of the structure of domains, a further limitation of AlphaFold we experienced here (and with other modular proteins) was the correct positioning of the respective modules. In the case of DpoMK34, the first 89 amino acids (which form the conserved N-terminal domain) appear to be folded inwards, in relation to the rest of the structure (Figure 3). This is not something we would have expected, based on its resemblance to the T7 tail fiber. However, AlphaFold’s output includes confidence measures for the position of each amino acid in the predicted structure relative to all other amino acids (Figure 4). These so-called predicted alignment errors show high predicted errors for every of the three N-terminal ends in the trimeric structure, relative to the rest of the structure (indicated by the red bands on the heat map). The high predicted errors show that AlphaFold is not certain about the position of those three N-terminal ends with respect to the position of the rest of the structure (Figure 4).

BLASTp analysis of DpoMK34 shows a low identity with tailspikes of phages having genomes highly similar to the genome of phage PMK34 (MN433707), including Acinetobacter phage vB_AbaP_APK81 (96.45% genome similarity; QNO11418.1), Acinetobacter phage IME-200 (96.12% genome similarity; YP_009216489.1), Acinetobacter phage vB_AbaP_AS12 (90.59% genome similarity; YP_009599229.1), Acinetobacter phage vB_AbaP_AS11 (82.13% genome similarity; YP_009599281.1) and Acinetobacter phage vB_ApiP_P1 (87% genome similarity; YP_009610482.1). More specifically, only the N-terminal anchor domains are conserved among those phages, whereas the C-terminal domains are divergent. On the other hand, the C-terminal domains of DpoMK34 are present in other phage tailspikes or tail fibers with a different N-terminal domain (e.g., predicted tail fiber of Acinetobacter phage SH-Ab 15599 belonging to the *Ackermannviridae* family; AXF41547.1). The chimeric *orf45* thus appears to result from a horizontal transfer event. Such mosaic evolution is a hallmark feature of the evolution of tailspikes and tail fibers [26].

The function of the most distal domain in the C-terminus is not yet known. Seven phage tailspike proteins (originating from phages: vB_AbaP_WCHABP5, vB_AbaP_B3, vB_AbaP_WU2001, vB_AbaP_D2, vB_AbaP_D2M, phiAB6 and SWH-Ab-3 corresponding to accession numbers: YP_009604582.1, YP_009610379.1, QVQ34730.1, YP_009624618.1, QFG15400.1, YP_009288671.1, YP_009949108.1, respectively) with at least 94% identity over the whole length of the protein were found. Remarkably, a short tetrapeptide (AAGT, aa 681-684 in DpoMK34) with high diversity was identified in a highly conserved region of the C-terminal domain (Figure 5A). A consensus sequence S/A–S/A/R–N/G–A/T was established. Strikingly, this variable region is located in the most distal, protruding loop of the protein (Figure 5B). This position hints at a biological role in the very early interaction with the host and is reminiscent of the corresponding ultimate loops of the tail fiber of *Escherichia coli* podophage T3 (T3gp17) [36]. These loops are coined host-determining regions and have been shown to be involved in tail fiber specificity targeting the O-antigen. The host-determining regions of T3gp17 were found to function similar to the complementarity-determining regions involved in antigen recognition of antibodies. In addition, Yehl et al. [36] showed that T3 phages change their host specificity by mutations in these particular loops of T3gp17, representing hotspots of evolution to counteract resistance development.

### 2.2. DpoMK34 Is a Specific Enzyme with Mesophilic Properties

For experimental validation of the predicted depolymerase activity of DpoMK34, *orf45* was cloned into expression vector pVTD3 [37], which introduces a C-terminal 6 × His. Recombinant DpoMK34 was produced in *E. coli* BL21(DE3) CodonPlus-RIL cells and purified by nickel affinity chromatography to electrophoretic homogeneity with a yield of 1 mg/L expression culture (Figure 6A). Volumes of 10 µL purified protein (or PBS as control) from a two-fold dilution series starting from 1.32 µM (100 µg/mL) were spotted onto soft agar seeded with *A. baumannii* MK34 cells in the exponential growth phase. After 16 h at 35 °C, zones with a reduced turbidity were observed starting from 0.08 µM (6.25 µg/mL) DpoMK34, indicative for depolymerase activity (Figure 6B). The spots are not fully cleared (as would be the case for example when dropping a phage lysin, lysing and thus killing the cells) but shows reduced turbidity (similar to the reduced turbidity of halos growing around phage plaques). This loss in turbidity is a hallmark feature for all depolymerase linked to capsule loss. No turbidity reduction was seen on plates with a multidrug-resistant strain (*A. baumannii* RUH134) and two colistin-resistant (*A. baumannii* Greek46, Greek47) strains that were included as additional controls (data not shown). Equal agar blocks (4 mm × 4 mm) were excised from the spots dropped with PBS, 1.32 µM (100 µg/mL) and 0.08 µM (6.25 µg/mL) DpoMK34 for bacterial cell counting. No significant difference (*p* > 0.05) in bacterial count was observed between the different treatments, excluding any antibacterial activity of DpoMK34 against *A. baumannii* MK34 (Figure 6C). For this small panel, the specificity of DpoMK34 thus corresponds to the narrow host spectrum of phage PMK34, which has been reported previously [25]. The capsular serotype of *A. baumannii* MK34 is not known. The high level of identity (97%) between DpoMK34 and the tailspike of Acinetobacter phage B3 suggests capsular serotype K2 [11] as the target for DpoMK34, but the divergence in the tetrapeptide of DpoMK34 (aa 681-684; AAGT) and the tailpike of Acinetobacter phage B3 (SSNA) may rather suggest a different host specificity, given the role of the hypervariable loop in host specificity in the case of phage T3. We show here that DpoMK34 and phage PMK34 are active on *A. baumannii* CIP 110467 having the K2 serotype, confirming that (at least) the K2 serotype is susceptible for the action of DpoMK34, thereby excluding a role for the divergent tetrapeptide in capsular serotype recognition.

A semi-quantitative assay was conducted to evaluate the thermal and pH stability of DpoMK34. The minimum halo-forming concentration (MHFC) of DpoMK34 was 0.08 µM (6.25 µg/mL) for DpoMK34 incubated at 4 °C. Elevating incubation temperature to 30, 40, and 50 °C for 30 min increases the MHFC to 0.16 (12.5 µg/mL), 0.33 (25 µg/mL) and 0.66 µM (50 µg/mL), respectively (Figure 7A). However, no activity was observed up to 1.32 µM (100 µg/mL, at least 16-fold increase) after incubation at higher temperatures (60 and 70 °C). Accordingly, we conclude that DpoMK34 is a typical mesophilic enzyme that retains approximately 10% of its activity at 50 °C. DpoMK34 is stable within a pH range from 6 to 9 (MHFC = 0.08 µM, 6.25 µg/mL). Reduced activity is observed after exposure to buffer with pH 4 and 5 (MHFC = 0.16 µM, 12.5 µg/mL) and pH 10 and 11 (MHFC = 0.33 µM, 25 µg/mL). Enzymatic activity is completely abolished after exposure to extreme acidic (pH 3) and alkaline (pH 12) buffers (Figure 7B).

### 2.3. DpoMK34 Is a Primary Determinant for Phage PMK34 Adsorption and Responsible for Its Specificity

Phage PMK34 shows a fast adsorption of approximately 95% of the phage particles within 5 min when incubated with untreated *A. baumannii* MK34. However, its adsorption capacity significantly (*p* < 0.01) drops to 9% after 5 min when the *A. baumannii* MK34 cells are pre-incubated (2 h) with 1.32 µM (100 µg/mL) DpoMK34 and subsequently washed (Figure 8). The maximum adsorption is 16% of the phage particles after 20 min. These findings further corroborate that strain-specific CPS serves as a primary receptor for DpoMK34 of phage PMK34. Preceding removal of the capsule thus limits the capacity to adsorb to the cell surface.

### 2.4. Decapsulating A. baumannii MK34 Cells with DpoMK34 Does Not Sensitize Them to the Action of Imipenem, Amikacin, and Colistin

Exopolysaccharide has been reported as barrier that impedes penetration of different antibiotics, which could mediate resistance [38,39]. Therefore, we evaluated whether DpoMK34 pre-treatment or co-treatment of *A. baumannii* MK34 with DpoMK34 could improve susceptibility to different antibiotics. For this goal, we selected imipenem (cell wall biosynthesis inhibitor), amikacin (protein synthesis inhibitor), and colistin (outer membrane permeabilizer and inner membrane disruptor) based on their different cellular targets [40]. As expected, DpoMK34 alone did not exhibit inhibitory activity [minimum inhibitory concentration (MIC) > 6.6 µM equivalent to 500 µg/mL]. The MIC values of imipenem, amikacin, and colistin against *A. baumannii* MK34 are 107 µM (32 µg/mL), 874 µM (512 µg/mL) and 0.87 µM (1 µg/mL), respectively, classifying the MK34 strain as resistant to both imipenem and amikacin, while sensitive to colistin. Neither pre-treatment nor co-treatment of *A. baumannii* MK34 with DpoMK34 (up to 6.6 µM equivalent to 500 µg/mL) changed the MIC value of the tested antibiotics. DpoMK34 thus does not improve the response of *A. baumannii* MK34 to the tested antibiotics.

### 2.5. DpoMK34 Sensitizes A. baumannii MK34 to Serum Killing

CPS serves as a virulence factor that helps the pathogen to escape from phagocytosis and complement-mediated killing. Hence, removal of the bacterial capsule of strain MK34 may sensitize bacteria to immune components of human serum. In this regard, we investigated the effect of different serum concentrations on DpoMK34-treated *A. baumannii* MK34 cells (Figure 9). Pretreated cells (1.32 µM DpoMK34 equivalent to 100 µg/mL; 2 h) show a 1.8 ± 0.34 log unit reduction in 25% (*v*/*v*) serum compared to PBS treated cells, which were resistant. This reduction further increases to a maximal reduction of 5.05 log units (below detection limit) in 50% (*v*/*v*) human serum, which is also observed in 75% (*v*/*v*) human serum. When DpoMK34-treated *A. baumannii* MK34 cells are exposed to heat-inactivated serum, no bacterial cell number reductions were observed up to the highest tested serum concentration [75% (*v*/*v*)]. This highlights the role of serum complement to kill DpoMK34-treated *A. baumannii* MK34.

### 2.6. Phage Resistant Mutants Are Insusceptible for DpoMK34, but DpoMK34 Exposure Does Not Easily Trigger Resistance

Emergence of phage resistant mutants is a common observation upon phage exposure, and we made a similar observation upon incubation of phage PMK34 with its host strain [25]. Spotting DpoMK34 (1.32 µM equivalent to 100 µg/mL) and phage PMK34 on lawns of three phage resistant clones that previously emerged within a plaque, did not yield zones of reduced turbidity or plaques, demonstrating insensitivity of the mutants to DpoMK34 and phage PMK34. An attempt to induce a similar phenotype by repeated exposure of the same wildtype strain to purified DpoMK34 did not easily yield insensitive mutants. Specifically, cells from a DpoMK34 spot-on-lawn were picked and incubated in LB broth supplemented with either DpoMK34 (final concentration of 1.32 µM equivalent to 100 µg/mL) or PBS for three consecutive sub-cultures (24 h for each). Then, the serially exposed cells were subjected to 1.32 µM DpoMK34 and phage PMK34 in a new spot-on-lawn assay. The cells remained sensitive to both DpoMK34 and phage PMK34 (data not shown).

## 3. Discussion

We have previously isolated lytic phage PMK34 that specifically infects the extensively drug-resistant *A. baumannii* strain MK34 [25]. Phage PMK34 plating results in clear lysis plaques with surrounding turbid zones (Figure 1), which are commonly observed for phages equipped with depolymerases [11,12,13,14,15,16,28]. In this study, we have identified and characterized the phage PMK34 capsule depolymerase DpoMK34 encoded by *orf45*.

Depolymerases are mostly present as part of phage tail structures (tail fiber, tailspike or baseplate) as reported for phages infecting *A. baumannii* [11,12,15], *E. coli* [17,41], and *Klebsiella* spp. [14,28]. However, another study described a Dpo7 depolymerase with a pectate lyase domain being part of the vB_SepiS-phiIPLA7 capsid as a predicted pre-neck appendage [42]. The conserved N-terminal domain of DpoMK34 is highly similar to its equivalent in other homologous phages (Figure 2), whereas the similarity to the N-terminus of the T7 tail fiber (gp17) is lower. Nonetheless, the classification of PMK34 as member of the *Autographiviridae* suggests that DpoMK34 is present as a tailspike anchored by its N-terminus similar to the canonical *Autographiviridae* phage T7. On the other hand, variation of pectate lyase domains among the aligned depolymerases (Figure 2) along with different host spectra of their parental phages corroborates the role of this central domain in determining the host spectrum based on different capsule serotypes or other surface receptors. More than 100 capsular serotypes have been identified in *A. baumannii*, explaining the tremendous variation among pectate lyase domains in *A. baumannii* phages as a result of phage–host co-evolution [5]. In the current study, we followed a sequential in silico and experimental analysis to confirm both identity and activity of DpoMK34 as a depolymerase. BlastP analysis revealed putative depolymerases with query coverage up to 100% and similarities >95%, but with a hypervariable 4 aa hotspot in the most ultimate loop. Structure modelling predicted an elongated protein structure with a typical right-handed *β*-helical topology (Figure 2), suitable for capsule puncturing. Moreover, this shape allows better scanning and recognition of complementary receptors expressed on the host surface [43]. The initial in silico annotation of DpoMK34 as a putative depolymerase is confirmed by observing zones of reduced turbidity (not clear zones) upon spotting purified DpoMK34 on soft agar seeded with *A. baumannii* MK34 (Figure 6). The resulting zones have a reduced turbidity similar to the turbidity of halos surrounding Acinetobacter phage PMK34 plaques (Figure 1B). Bacterial cell counting in the spots showed the absence of bacterial killing as would happen when the cells are lysed by a lysin.

DpoMK34 has a specific polysaccharide depolymerase activity against *A. baumannii* MK34 (which is the host strain for phage PMK34) within submicromolar ranges (0.08 µM equivalent to 6.25 µg/mL, Figure 6) and is also active against strain CIP110467 with a K2 capsular serotype (data not shown). This highlights the differences in specificity determinants between a tail fiber that binds its receptor and a tailspike with enzymatic activity. Indeed, whereas the specificity of the T3gp17 tail fiber binding to the O-antigen could be explained by these distal loops [36], other determinants are responsible for the capsular serotype specificity of tailspikes with depolymerase activity. Residues located in cavities and groves located on intra- or intersubunit surfaces of the trimeric tailspikes have been reported to be involved in serotype specificity [44,45]. Therefore, we speculate that the highly divergent tetrapeptide might be involved in binding a possible secondary receptor (e.g., LPS, in analogy to T3gp17). Generally, the function of the C-terminal domain in depolymerases remains largely understood, except when encoding a chaperone domain needed for folding and trimerization [28]. Further investigation is required to understand the highly divergent loop in the case of tailspikes with depolymerase activity.

Furthermore, DpoMK34 is stable across a wide range of temperatures (4 to 50 °C) and pH values (4 to 11, Figure 7). Such robustness is consistent with the function of tailspikes as structural proteins, naturally evolved to work from without under different temperatures and pH values [11,14,15,17,28]. Based on the coinciding and limited spectrum of bacteriophage PMK34 and its depolymerase (DpoMK34) and the disappearance of the capsule in cells within the halo zone (Figure 1), we conclude that phage PMK34 utilizes a specific capsule serotype as a receptor to initiate infection. This hypothesis was supported by a significant drop (*p* < 0.01) in the percentage of adsorbed phage PMK34 particles from >95% to 9% after treatment of bacterial cells with DpoMK34 (Figure 8). Diminished adsorption of parental phages after exposing bacterial hosts to respective recombinant depolymerases has been reported for a few phages. For example, treating *A. baumannii* NIPH 2061 (K2 capsular serotype) with K2 depolymerase significantly decreased percentage of adsorbed *A. baumannii* phage B3 particles from 100 to less than 20% [11]. Similarly, phage B9 adsorption dropped to 20% when host bacteria were pre-treated with specific K30 and K45 depolymerases [15]. The observed remaining adsorption may be explained by a recovery of CPS, or a less effective adsorption process on exopolysaccharide remnants that remained after a preceding DpoMK34 treatment.

BlastP analysis revealed more than ten homologous putative depolymerases deposited at NCBI (Figure 2). Three of them (K2, Dpo48, φAB6 tailspike protein) have been assessed in terms of thermal and pH dependency [11,12]. For example, K2 and Dpo48 depolymerases were found to have a similar pH activity range of 5–9, being a more narrow pH range than that of DpoMK34. On the other hand, both depolymerases (K2 and Dpo48) showed higher thermal stability with retained activity up to 70 °C than the DpoMK34 and φAB6 tailspike proteins (activity retained up to ~50 °C). The thermal stability of DpoMK34 was evaluated using a semi-quantitative spot-on-lawn assay while thermal stability of the others (K2, Dpo48, phiAB6 tailspike protein) was assessed using the 3, 5-dinitrosalicylic acid assay measuring the release of reducing sugars.

The capsule depolymerase from *Klebsiella pneumoniae* phage KP36 (DepoKP36) shows no synergy with oxytetracycline, chloramphenicol, or ciprofloxacin against *K. pneumoniae* [14], whereas another *Klebsiella*-specific phage depolymerase is synergistic with polymyxin when treating biofilms [24]. In addition, a bacterial depolymerase isolated from *Aeromonas punctate* is synergistic in vivo with gentamycin in a murine infection model [23]. To our knowledge, the synergy between antibiotics and a depolymerase was not yet evaluated against *A. baumannii*. In this study, we have selected different antibiotics with different molecular targets located in different cellular sites. The antibiotics tested and DpoMK34 were neither synergistic nor antagonistic. These findings indicate that the CPS of *A. baumannii* MK34 does not affect the response of the host strain to the tested antibiotics and are in line with the in vitro findings of DepoKP36. It remains to be investigated whether the bacterial *Aeromonas* origin of the depolymerase, the in vivo evaluation, biofilm mode or the specific antibiotic explains the previously observed synergies.

CPS is an important virulence factor that helps bacteria to evade host immunity. Accordingly, sensitization to serum killing upon removal of the CPS has previously been reported for different pathogens including *A. baumannii*, *K. pneumoniae,* and *Providencia stuartii*, following an apparently similar mechanism [11,12,14,16]. Moreover, depolymerization of the O-antigen polysaccharide by a phage tail-associated depolymerase sensitizes *E. coli* to serum [17]. DpoMK34 sensitizes *A. baumannii* MK34 to serum killing in a serum concentration-dependent manner (Figure 9) with a full eradication (5 log unit reductions, detection limit) from 50% (*v*/*v*) serum concentration. This effect was completely abolished after heat inactivation of serum complement, highlighting the role of the capsule polysaccharide in shielding off the access to serum complement. This is further corroborated by a previous report comparing the effect of both intact and heat-inactivated serum on *A. baumanni* strains pre-treated with specific depolymerases [11,12]. For instance, stationary growing bacterial cells treated with Dpo48 showed a 3 log unit reduction in bacterial count after mixing with 25% (*v*/*v*) serum, whereas a 5 log unit reduction was obtained when the serum proportion increased to 50% (*v*/*v*). Serum sensitizing was completely absent when stripped bacterial cells were exposed to complement-inactivated serum [12]. Similarly, *A. baumannii* treated with K2 depolymerase shows a serum-dependent bacterial count reduction (below detection limit), which is reversed when complement was inactivated [12].

DpoMK34 (and other phage-encoded depolymerases) can thus be classified as an antivirulence compound. Antivirulence compounds reduce or eliminate the pathogen’s virulence, which can be sufficient to eliminate the infection of the disarmed pathogens by a potent immune system. Since such compounds do not exert killing of the bacterial cells, the selection pressure to develop resistance is believed to be significantly reduced [46]. This agrees with our observations for DpoMK34. Whereas resistance development can be easily found for the parental phage PMK34 after single exposure (with co-resistance to DpoMK34), serially DpoMK34-exposed cells retained their susceptibility to both DpoMK34 and phage PMK34, which is in line with previous reports about depolymerases targeting different pathogens [11,16,17]. Phage resistance can occur by a large diversity of mechanisms (reviewed in [47]) and was also found for phage PMK34 [25]. The most plausible mechanism of phage resistance observed in this work is capsule loss or modification of its composition, causing the loss of the primary phage receptor, and consistently also caused resistance to DpoMK34. The latter resistance mechanism was not induced by the DpoMK34 itself in our simple in vitro setup. Either the capsule integrity was insufficiently affected (e.g., impartial removal of the capsule, regain of a capsule during prolonged exposures) or the selection pressure was too low because capsule loss does not result in a fitness cost under laboratory conditions with rich nutrients. It is unclear whether this finding will hold true in vivo. There, the mode-of-action of DpoMK34 may effectively result in cell killing and clearance in concert with the immune system. The selection pressure in this environment is thus different. Loss of the capsule comes at the cost of its virulence, but if the strain is able to modify its capsule polysaccharide composition without losing its virulence, it will be positively selected. Analogously, resistance mechanisms against other antivirulence compounds have been reported as well [48,49,50]. The narrow specificity of DpoMK34 and other phage-encoded depolymerases is thus their Achilles heel. Therapeutic application would necessitate a large arsenal of depolymerases to make cocktails similar to the approach of phage cocktails, or a depolymerase with broadened specificity produced by protein engineering, if possible.

In conclusion, depolymerases constitute an alternative approach that could be used for control of antibiotic unresponsive pathogens. Their ability to support the immune system to eliminate virulent pathogens with reduced probability of selecting resistant mutants are main features that set them apart from other antibacterials. DpoMK34 of Acinetobacter phage vB_AbaP_PMK34 has been extensively characterized as a specific depolymerase (K2 serotype). Whereas this high specificity limits collateral damage, it will also complicate broad application, necessitating cocktail compositions or protein engineering to broaden the specificity. At the structural level, DpoMK34 is predicted to adopt a fibrous structure as in a tail fiber (although shorter than in T7), but with a typical β-helix and depolymerase activity, usually associated with tailspikes. In addition, DpoMK34 has an unexpected hypervariable ultimate loop which at least does not affect K2 serotype specificity.

## 4. Materials and Methods

### 4.1. Phage, Bacterial Strains, Culturing Conditions, and Capsular Staining

*E. coli* BL21(DE3) CodonPlus-RIL cells (Agilent Technologies, Inc., Santa Clara, California, USA) were used for protein expression. *Acinetobacter baumannii* strains RUH134 [51], MK34, Greek46, Greek47 [25], and CIP 110467 (K2) were used to analyze the specificity of the purified depolymerase. All strains were grown in LB broth [1% (*w*/*v*) tryptone, 0.5% (*w*/*v*) yeast extract and 1% (*w*/*v*) NaCl] at 37 °C while shaking (250 rpm), or on LB agar plates (LB supplemented with 2% agar). Kanamycin (50 µg/mL), chloramphenicol (25 µg/mL), and sucrose 5% (*w*/*v*) were added to LB broth or agar for selection of the transformed *E. coli* cells. Acinetobacter phage vB_AbaP_PMK34 (phage PMK34) was previously isolated, characterized, and sequenced. A double layer assay [52] was used to produce plaques against overnight culture of *A. baumannii* MK34 using phage PMK34 diluted to (10^2^ PFU/mL) in SM buffer (10 mM Tris-HCl, 10 mM MgSO_4_, and 100 mM NaCl, pH 7.5). The plates with the plaques were further incubated for additional 24 h at 30 °C to observe the expansion of the halo zones around the plaques. Negative staining of the capsular polysaccharide surrounding bacterial cells inside and outside halo zones was performed using Maneval’s solution [5% (*v*/*v*) acetic acid, 4% (*v*/*v*) phenol, 5% (*w*/*v*) FeCl_3_·6H_2_O and 0.05% (*w*/*v*) acid fuchsin] as described before [53]. The stained cells were then visualized using light microscopy with an oil immersion lens (100×) (Leica Microsystems, Wetzlar, Germany). The phage genomic sequence was deposited in NCBI gene bank under accession number MN433707 [23]. The accession number of DpoMK34 is QGF20174.1.

### 4.2. In Silico Analysis of orf45

Previous phage PMK34 genome annotation and BLASTx analysis [54] showed a putative tailspike (*orf45*) with a central pectate lyase conserved domain among the lately expressed proteins preceding the host lysis module [25]. For further bioinformatic analysis of the putative tailspike, we carried out conserved domain analysis using the Protein family database (Pfam) online server [55]. Moreover, amino acid sequence alignment of the putative depolymerase (DpoMK34) with other homologous proteins was carried out using Geneious^®^ 7.1.3 software (https://www.geneious.com, accessed on 20 January 2021). Isoelectric point (pI) and the theoretical molecular weight were predicted using the ProtParam tool of the Expasy server (https://web.expasy.org/protparam/, accessed on 20 January 2021) [56]. The trimeric structure was predicted using AlphaFold-Multimer using Ghent University’s supercomputing resources (more specifically, two NVIDIA Volta V100 GPUs with 32GB graphics processing unit (GPU) memory) [57]. Predicted Local Distance Difference Test (LDDT) scores and alignment errors were visualized using code from ColabFold [58]. BlastP (protein–protein Blast) of the DpoMK34 amino acid sequence was performed against the non-redundant protein sequences (nr) database using standard parameters (expect threshold: 0.05, word size: 6, MATRIX: BLOSUM62, Gap cost: existence 11, extension 1, conditional compositional score matrix adjustment). Hits with 100% query coverage and at least 94% identity (E-value 0.0) were selected for multiple alignment with expected value better than 0.001. Multiple alignment of query and target sequences was visualized with WebLogo, the online available tool [59].

### 4.3. Cloning, Expression, and Purification of the Putative Depolymerase

Amplification of the putative depolymerase coding sequence (*orf45*) was carried out using specific primers (forward 5′-GTC**GGTCTC**ACCATGAATATACTACGCTCATTTACAGAGACAGTGG-3′ and reverse 5′-GTC**GGTCTC**ATACTTACTCGTTGCTGTAAATGC-3′; IDT, Leuven, Belgium) with *Bsa*I restriction sites (bold) using the phage PMK34 genome as template and PHUSION DNA polymerase (Thermo Fisher Scientific, Waltham, MA, USA). Afterwards, the amplicon was purified, digested using 1 U of *Bsa*I (Thermo Fisher Scientific, Waltham, MA, USA), and introduced into expression vector pVTD3 [37] using T4 DNA ligase (Thermo Fisher Scientific, Waltham, MA, USA) to add a 6×His tag at the C-terminus [37]. Heat shock was carried out to transform chemical competent *E. coli* BL21(DE3) CodonPlus-RIL cells (Agilent Technologies, Inc, Santa Clara, CA, USA) with pVTD3/*orf45*. Isolated recombinant plasmid was validated using Sanger sequencing (LGC, Berlin, Germany). Recombinant *E. coli* cells were then grown up to the exponential phase (OD_655nm_ = 0.6) in LB broth supplemented with 50 µg/mL kanamycin, followed by addition of 0.5 mM of isopropyl-^®^-D-thiogalactopyranoside (IPTG; Thermo Fisher Scientific, Waltham, MA, USA) and 18 h incubation at 16 °C. The cells were then collected (4000× *g* for 30 min) and lysed as described before to release His-tagged DpoMK34 [37]. Purification was performed using His GraviTrap™ columns (GE Healthcare, Chicago, IL, USA) using the manufacturer’s instructions. Finally, the purified protein was dialyzed against PBS pH 7.4 (137 mM NaCl, 2.7 mM KCl, 10 mM Na_2_HPO_4_, and 1.8 mM KH_2_PO_4_) using 3.5 K molecular weight-cutoff (MWCO) Slide-A-Lyzer MINI Dialysis Devices (Thermo Fisher Scientific, Waltham, MA, USA) and sterilized using a 0.22 µM filter (PES, Novolab, Geraardsbergen, Belgium). Protein purity was assessed using 12 (*w*/*v*) % sodium dodecyl sulfate–polyacrylamide gel electrophoresis (SDS-PAGE) whereas its quantification was performed by UV measurement at 280 nm with a DS-11 spectrophotometer (DeNovix Inc., Wilmington, DE, USA).

### 4.4. DpoMK34 Enzymatic Activity and Stability Assessment

The enzymatic activity of DpoMK34 was analyzed using a spot-on-lawn assay as described before [14]. Hundred microliters of exponentially growing (OD_655nm_ = 0.6) cells of different *A. baumannii* strains (MK34, RUH134, Greek46, Greek47) were mixed with 4 mL LB soft agar [0.5% (*w*/*v*)] and then spread over the surface of plain LB agar [2% (*w*/*v*)]. After solidification, 10 µL of a two-fold serial dilution of DpoMK34 (1.32–0.04 µM, equivalent to 100–3.1 µg/mL) was spotted on a bacterial lawn, left for air drying and incubated for 16 h at 35 °C. A positive result was visible as a halo zone of reduced turbidity. PBS pH 7.4 (10 µL) was spotted as a control. The minimum halo-forming concentration (MHFC) was defined as the lowest concentration that causes a zone of reduced turbidity visible with the naked eye. Bacterial cell counts for zones spotted with 100 µg/mL and 6.25 µg/mL of purified DpoMK34 were performed by suspending excised agar blocks (4 × 4 mm) from the treated surfaces, or PBS as control, in 2 mL of PBS pH 7.4 followed by vigorous vortexing for 1 min. Afterwards, the bacterial suspensions were serially diluted in the same buffer and plated for counting.

Thermal stability was assessed by incubating 2.64 µM (200 µg/mL) DpoMK34 at different temperatures (4, 30, 40, 50, 60, and 70 °C) using a water bath for 30 min, followed by cooling down on ice during 10 min. A two-fold dilution series of the incubated protein solutions were then spotted as described above. Similarly, pH stability was evaluated by dialyzing stock solution of DpoMK34 (2.64 µM) against Britton Robinson universal buffer (0.04 M H_3_PO_4_, 0.04 M H_3_BO_3_, 0.04 M CH_3_COOH, and 0.15 M NaCl) adjusted to the pH test range (4–11). After 30 min incubation, dilutions were prepared using the same universal buffer with the respective pH and spotted on a bacterial lawn. A drop (10 µL) of the universal buffer with the respective pH was used as control.

### 4.5. Phage Adsorption Assay

A phage adsorption assay was carried out as described previously [16] using both wild and stripped *A. baumannii* MK34 phenotypes. Briefly, exponentially grown (OD_655nm_ = 0.6) *A. baumannii* MK34 cells were resuspended in either PBS pH 7.4 or DpoMK34 (final concentration of 1.32 µM, equivalent to 100 µg/mL). After two hours incubation at 35 °C, cells were pelleted (16,000× *g* for 5 min) and washed twice using LB broth. The cells were then diluted to 10^6^ CFU/mL with LB broth and incubated with phage PMK34 at multiplicity of infection (MOI) of 0.01 and incubated at 35 °C. At specific time intervals (5, 10, and 20 min) 100 µL samples were added to 900 µL cold LB and spun down (16,000× *g* for 5 min). The number of non-adsorbed phage particles was counted using a double layer assay [48]. Then, the number of adsorbed phage particles was calculated by subtracting the count of non-adsorbed phages in the supernatants from the initial number (10^4^ PFU/mL). Finally, the percentage of adsorbed phage particles was calculated relative to the initial number (10^4^ PFU/mL).

### 4.6. MIC Detection and Checkerboard Analysis

The broth microdilution method in Mueller–Hinton broth (MHB) was used to measure minimum inhibitory concentrations (MICs) of DpoMK34 along with different antibiotics including imipenem, amikacin, and colistin within test ranges of 0–6.6 µM (0–500 µg/mL), 0–214 µM (0–64 µg/mL), 0–874 µM (0–512 µg/mL), and 0–56 µM (0–64 µg/mL), respectively [60]. This assay was performed with exponentially growing cells (OD_655nm_ = 0.6) of *A. baumannii* MK34 with final density of 10^6^ CFU/mL. Subsequently, we followed a checkerboard approach [61] to assess the combinatorial outcome of two-fold dilution series of DpoMK34 with each antibiotic at concentration ranges of 0–6.6 µM (0–500 µg/mL) for DpoMK34 and 0–2 × MIC for the antibiotic. In addition, we assessed the MIC value of each antibiotic against DpoMK34-treated *A. baumannii* MK34 cells. Therefore, exponentially grown *A. baumannii* cells (OD_655nm_ = 0.6) with final density of 10^6^ CFU/mL were incubated with either PBS pH 7.4 or 1.32 µM DpoMK34 (final concentration equivalent to 100 µg/mL) for 2 h at 35 °C. Afterwards, the cells were pelleted (16,000× *g* for 5 min), washed twice using Mueller–Hinton broth and used for MIC assays using the same antibiotic concentration ranges as mentioned above.

### 4.7. In Vitro Serum Sensitizing Assay

The ability of DpoMK34 to render serum-resistant *A. baumannii* MK34 prone to serum killing was assessed as described elsewhere [16]. Briefly, *A. baumannii* MK34 cells were grown to the exponential phase (OD_655nm_ = 0.6) in LB, spun down (16,000× *g* for 5 min), washed twice and diluted 100 times, to reach final bacterial count of 10^6^ CFU/mL using PBS pH 7.4. Subsequently, these cells were treated with DpoMK34 (final concentration 1.32 µM equivalent to 100 µg/mL) at 35 °C for 2 h. The treated cells were then mixed in different proportions with human serum (MP Biomedicals, Leuven , Belgium), including [1:3 (25% *v*/*v* serum), 1:1 (50% *v*/*v* serum), and 3:1 (75% *v*/*v* serum)]. These mixtures were then incubated at 37 °C for additional 2 h, serially diluted in 100% human serum and plated on LB agar plates for bacterial cell counting. PBS pH 7.4 treated *A. baumannii* MK34 cells were included as control. In parallel, a similar experiment was conducted using heat-inactivated serum (56 °C for 30 min).

### 4.8. Resistance Development Assay

Three phage PMK34 resistant mutants (colonies growing within plaques) were subcultured in LB for 24 h at 35 °C then dropped with 10 µL of 1.32 µM DpoMK34 or phage PMK34 (10^8^ PFU/mL) to assess their sensitivity. In addition, the possibility of resistance development after repeated exposures to DpoMK34 and phage PMK34 was investigated as described previously [11] with some modifications. In the case of DpoMK34, an overnight culture was prepared from *A. baumannii* MK34 cells picked up with a sterile loop from a zone of reduced turbidity in soft agar spotted with 10 µL of 1.32 µM DpoMK34 equivalent to 100 µg/mL). Subsequently, this overnight culture was 100-fold diluted in LB supplemented with either DpoMK34 (final concentration of 1.32 µM equivalent to 100 µg/mL) or PBS and incubated for 24 h at 35 °C. Afterwards, the cells were spun down (16,000× *g* for 5 min), washed twice with LB and sub-cultured two more times in LB with either DpoMK34 or PBS for 24 h at 35 °C. After the three rounds of exposure, the treated cells were spun down (16,000× *g* for 5 min), washed twice and allowed to grow in LB till reaching OD_655nm_ = 0.6. Subsequently, the sensitivity of the cells to DpoMK34 and phage PMK34 was evaluated using the same spot-on-lawn assay as described above.

### 4.9. Statistical Analysis

Statistical significance between means of different groups was assessed by two-tailed Student’s *t*-test using SPSS Statistics for Windows V. 22.0 (IBM Corp., Endicott, New York USA) Unless otherwise stated, statistical significance was considered when *p* < 0.05.

## Figures and Tables

**Figure 1 antibiotics-11-00677-f001:**
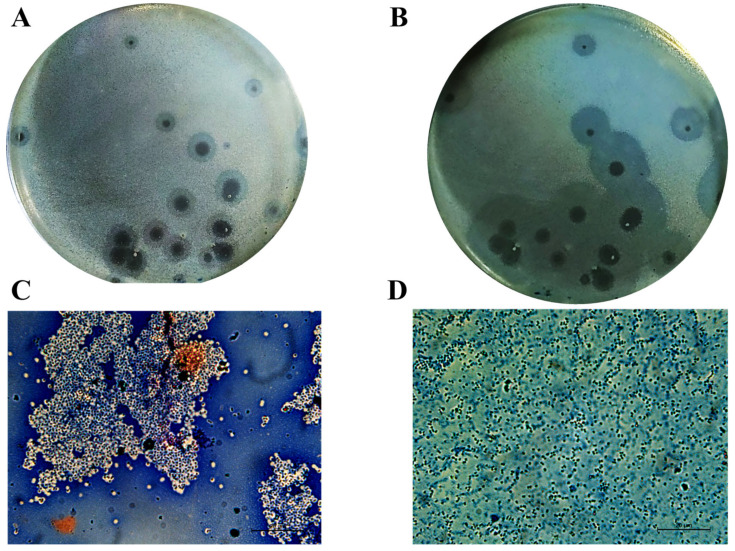
Cells in the turbid halo surrounding clear plaques lack their capsule. Acinetobacter phage PMK34 stock (10^11^ PFU/mL) was serially diluted in SM buffer pH 7.4 to a final concentration of 10^2^ PFU/mL. Then, 100 µL of the diluted phage was mixed with an equal volume of *A. baumannii* MK34 overnight culture in 4 mL soft agar overlaying LB agar. The plate was incubated at 30 °C for (**A**) 20 h and (**B**) 48 h. Growing halos surrounding each plaque are visible. Maneval’s staining (specific capsular negative staining) of (**C**) *A. baumannii* MK34 bacterial lawn and (**D**) cells inside the halo zone showed removal of the capsular polysaccharide layer (white shells surrounding dark rods) as visualized by light microscopy using an oil immersion lens (100×) (scale bar of 20 µm).

**Figure 2 antibiotics-11-00677-f002:**
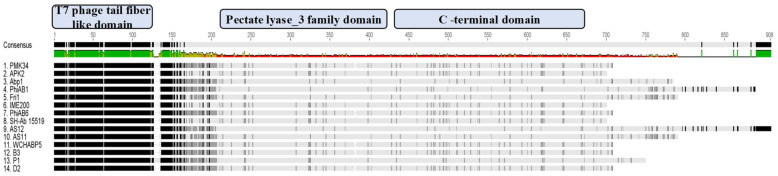
Multiple protein sequence alignment of DpoMK34 and tailspikes of highly homologous phages. Amino acid sequence alignment of phage PMK34 tailspike (DpoMK34) with tailspikes of homologous phages including APK2 (AZU99242.1), Abp1 (YP_008058239), phiAB1(YP_009189380.1), Fri1 (YP_009203055.1), IME200 (YP_009216489.1), phiAB6 (YP_009288671.1), SH-Ab 15519 (YP_009598268.1), AS12 (YP_009599229.1), AS11 (YP_009599281.1), WCHABP5 (YP_009604582.1), B3 (YP_009610379.1), P1 (YP_009610482.1) and D2 (YP_009624618.1) using Geneious^®^ 7.1.3. Per protein, the darkness per amino acid position indicates the level of similarity. The alignment shows a conserved N-terminal T7 phage tail fiber-like domain (amino acids 15 to 110), a variable central pectate lyase domain (amino acids 225 to 433) and variable C-terminal domain (amino acids 434 to 699) for DpoMK34. The amino acid consensus is shown on top with the level of conservation ranging from high to low, ranging from green to red, correspondingly.

**Figure 3 antibiotics-11-00677-f003:**
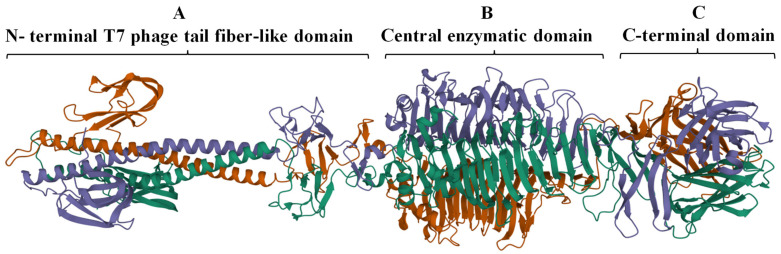
Predicted 3D model of DpoMK34 using AlphaFold-Multimer. The predicted DpoMK34 structure adopts an elongated trimeric protein with three distinct domains, including (**A**) an N-terminal anchor domain for attachment to the phage tail, connected via a long α-helix to (**B**) a central pectate lyase for enzymatic activity and (**C**) a C-terminal domain. The visualization was made with Mol* 3D (https://www.rcsb.org/3d-view, accessed on 4 March 2022). Different colors represent individual monomers composing the full structure of DpoMK34.

**Figure 4 antibiotics-11-00677-f004:**
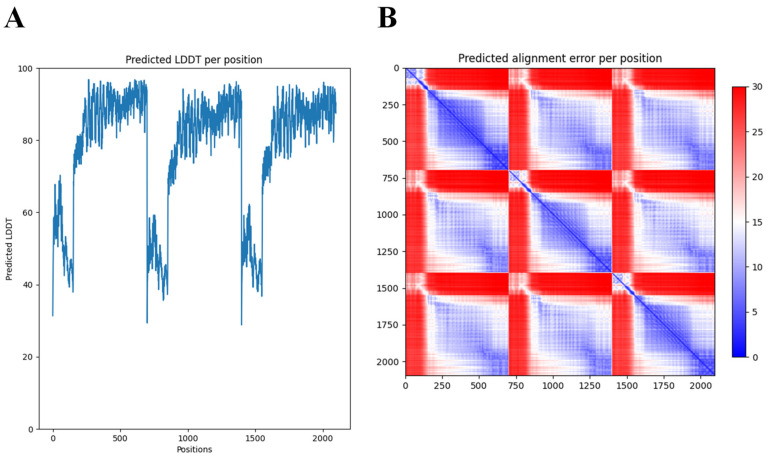
Predicted Local Distance Difference Test (LDDT) and alignment error per position of the 3D structure of DpoMK34 generated by AlphaFold-Multimer. (**A**) The predicted LDDT scores of the predicted trimeric structure by AlphaFold. These scores are a per-residue measure of how confident AlphaFold is about its prediction. Note that the position axis refers to the concatenation of the three monomers. (**B**) The predicted alignment error, which can be used to interpret the relative position of domains. A low predicted alignment error (blue) between the residues of different domains indicates that AlphaFold predicts the relative positions of these domains well. When the predicted alignment error between domains is high (red), the relative position of these domains is uncertain.

**Figure 5 antibiotics-11-00677-f005:**
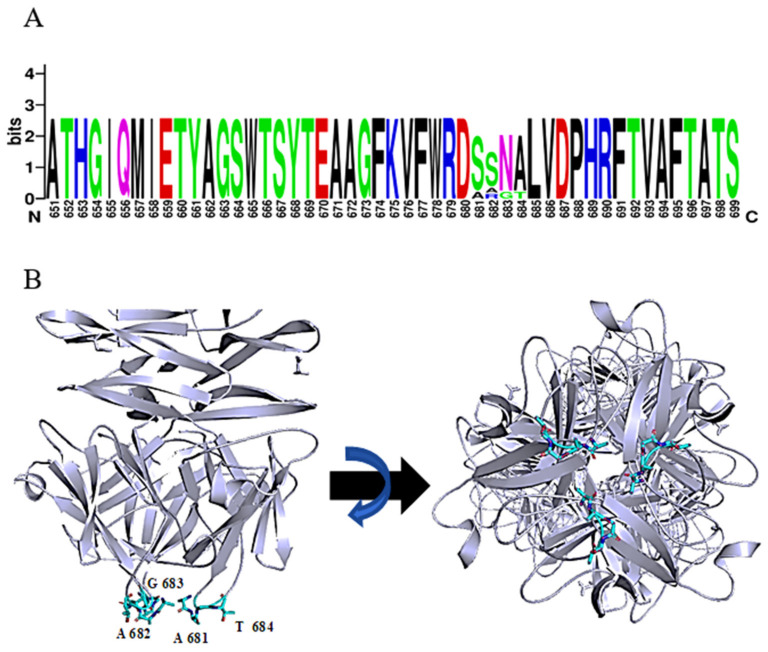
Consensus sequence S/A-S/A/R-N/G-A/T in the most ultimate loops of the C-terminal domain of DpoMK34. (**A**) Sequence identity with a variable region of 4 amino acids, visualized with WebLogo. (**B**) Predicted 3D model of DpoMK34 with the variable 4 amino acids located at C-terminal loop (Cyan colored residues; side and bottom views).

**Figure 6 antibiotics-11-00677-f006:**
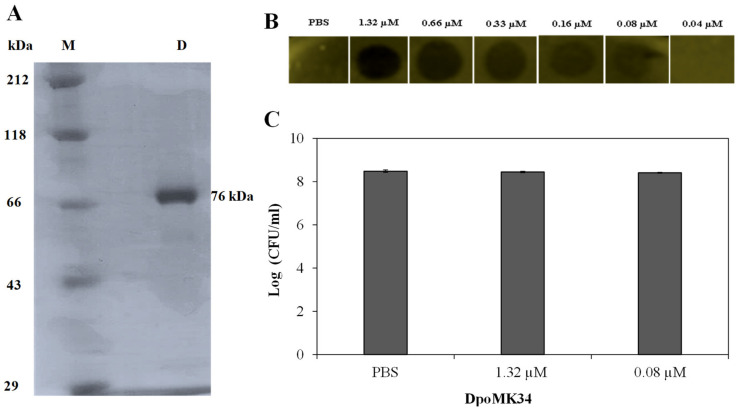
Activity of DpoMK34. (**A**) SDS-PAGE analysis of purified C-terminal 6×His-tagged DpoMK34. Lane (M) represents Roti^®^-Mark standard (Carl-Roth, Karlsruhe, Germany), whereas lane D represents purified DpoMK34. (**B**) Spot-on-lawn assay using different DpoMK34 concentrations on *A. baumannii* MK34 in the exponential growth phase. PBS pH 7.4 was used as control. (**C**) Bacterial counts from 4 × 4 mm excised squares treated with PBS, 1.32 µM DpoMK34, and 0.08 µM DpoMK34. Bars represent means ± standard deviation of three independent replicates.

**Figure 7 antibiotics-11-00677-f007:**
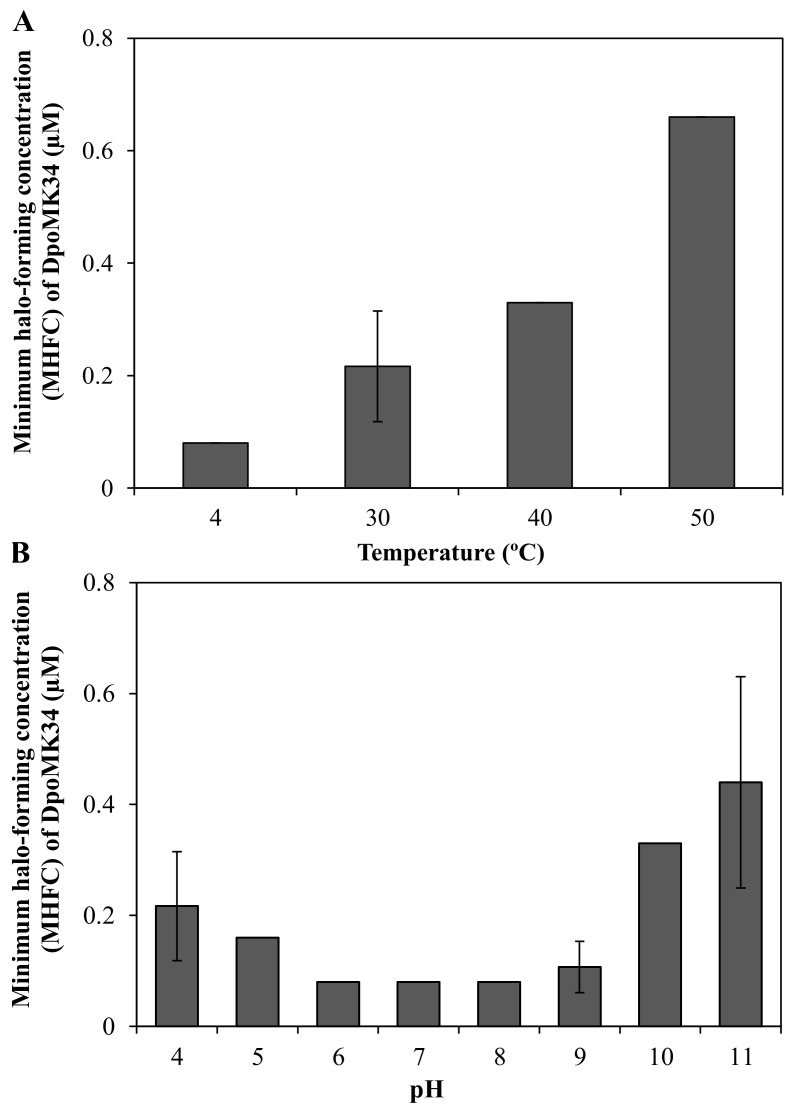
Semi-quantitative stability analysis of DpoMK34. (**A**) Thermal stability was assessed by incubating 2.64 µM (200 µg/mL) DpoMK34 at different temperatures (4–70 °C) for 30 min, followed by cooling down on ice for 10 min, two-fold serial dilution in PBS pH 7.4 and spotting on a lawn of *A. baumannii* MK34. The lowest concentration of DpoMK34 showing turbidity reduction was taken as the minimum halo-forming concentration. No activity was observed after incubation at 60 and 70 °C. (**B**). At temperature points of 4, 40, 50 °C the standard deviation equals zero. pH stability was analyzed by exchanging buffer of 2.64 µM (200 µg/mL) to universal buffer adjusted to different pH values ranging from 4 to 11, followed by further dilution in universal buffer with the respective pH and spotting on soft agar seeded with *A. baumannii* MK34 cells. At pH points of 5, 6, 7, 8, the standard deviation equals zero. Bars represent means ± standard deviation two independent experiments with three technical replicates.

**Figure 8 antibiotics-11-00677-f008:**
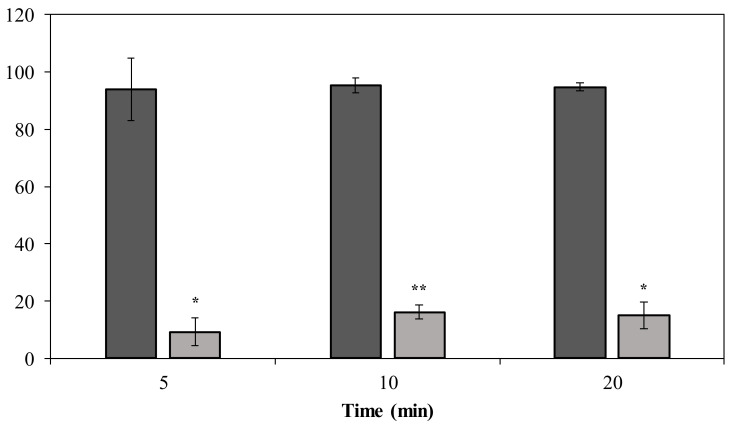
Phage MK34 adsorption after capsule removal. Intact (dark grey) and 1.32 µM DpoMK34 treated (light grey) *A. baumannii* MK34 cells were used for an adsorption assay by incubating exponentially growing cells with phage MK34 at MOI of 0.01. Bars represent means ± standard deviation of three independent replicates. Asterisks show statistical differences compared to phage MK34 adsorption to intact *A. baumannii* MK34 cells (Student’s *t*-test; * *p* < 0.01, ** *p* < 0.001).

**Figure 9 antibiotics-11-00677-f009:**
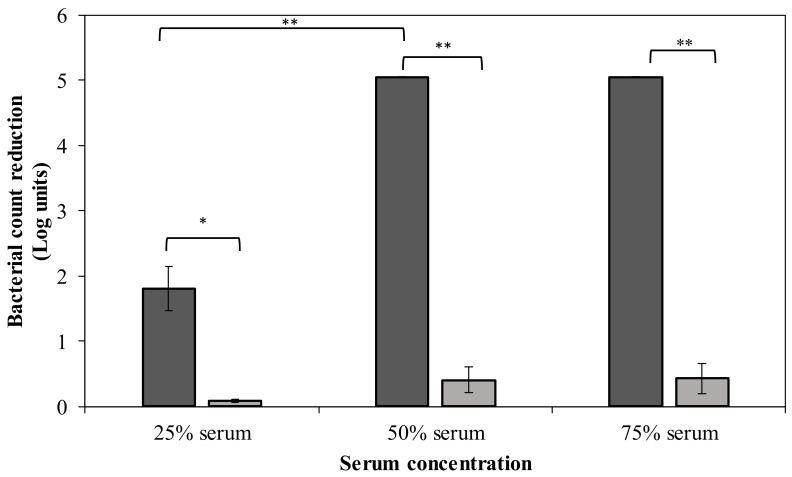
Sensitization to serum after DpoMK34 pretreatment. Exponentially growing *A. baumannii* MK34 cells are pelleted and diluted to 10^6^ CFU/mL using PBS pH 7.4. The cells were then treated with 100 µg/mL DpoMK34 for 2 h in 35 °C, washed and subjected to different serum concentrations [expressed in % (*v*/*v*)], followed by a 2 h incubation at 37 °C for 2 h. The bacterial cell count was subsequently determined. Both the effect of normal human serum (dark grey) and heat-inactivated serum (56 °C; 30 min) serum (light grey) were evaluated. Bars represent means ± standard deviation of three independent replicates. Asterisks show statistical differences compared to treatment in heat-inactivated serum, or between different serum concentrations (Student’s *t*-test; * *p* < 0.05, ** *p* < 0.01).

## Data Availability

The nucleotide sequence of Acinetobacter phage PMK34 and the amino acid sequence of its depolymerase were deposited in NCBI with accession numbers of MN433707 and QGF20174.1, respectively.

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
