# Peer review of "The Specific Capsule Depolymerase of Phage PMK34 Sensitizes Acinetobacter baumannii to Serum Killing"

_antibiotics, 2022, doi:10.3390/antibiotics11050677_

Round 1

Reviewer 1 Report

In the original article “The specific capsule depolymerase of phage PMK34 sensitizes Acinetobacter baumannii to serum killing” the authors have identified and analyzed a putative phage depolymerase encoding orf45, cloned, produced and purified the gene product and characterized the functions of this putative phage depolymerase.  

Main issues:

In the beginning the authors talk about putative depolymerase. But, quite soon, they start to use term depolymerase without questioning it. To me as a reviewer, it is still a bit vague if this enzyme is a depolymerase or a lysin or what? My doubts started with the Figure 6. In figure 6B the putative depolymerase causes clearance of the host bacteria on an agar plate but not in the control sample. Interestingly, the cell counts are similar in the control sample and in the two different depolymerase concentration samples. How is this possible? Is the reasoning behind that the depolymerase cleaves the capsule and leaves the bacteria alive. But why don’t these bacteria cause turbidity on the plate anymore if they are still alive? Or dead, since even dead cells would cause turbidity. Unfortunately, I don’t understand this, could you please clarify the reasoning behind this in the discussion so that the reader does not have to guess how did you ended up to your conclusions. In addition, please describe in the methods how you determined the cell counts, I could not find this. Also, please consider adding light microscopy images as in Fig 1. for these three different samples presented in 6C, maybe this would further clarify this issue.

If the authors think, there is enough evidence for this enzyme to be considered as a depolymerase, then this evidence - with appropriate references to the data and literature - should be discussed more in-depth and more clearly in the discussion so that no doubts are left for the reader. Or if there are other possible functions for this enzyme, these should be discussed in the discussion.

Another smaller, but still important, issue is that there are a lot of literature, figure and inner references missing in the text, please, be more careful with this. Otherwise, the text is well written and interesting.

Other comments:

Abstract:

Line 30: Please, add “A. baumannii” between clinical isolate and MK34.

Line 35: Please, check if there is extra space between words “receptor.” and “The”.

Line 38: Please, remove the spaces between numbers and %.

Introduction:

Later, you discuss about lipopolysaccharides (LPS; 2.1) and exopolysaccharides (EPS; 2.4). Could you discuss (shortly) the terminology and the differences between LPS, CPS and EPS in the introduction?

In the introduction, a reference to a good review article(s) about depolymerases would be also nice.

Line 49: Please, add a reference after the first sentence.

Lines 54 and 67: Please, add a reference.

Line 81: Please, use italics with orf45.

Results:

Line 88: Please, remove the dot from the end of the title.

Line 89: Please, replace “a” with “our”.

Lines 96-97: Was this work done in your previous publication (23)? If so, please refer to it.

Line 97: Please, use italics with orf45.

Line 99: Please, add an inner reference to the Materials and Methods 4.2 In silico analysis…

Line 101: Please, remove extra space after word “phages”.

Line 101: Please, add “s” after word “acid”.

Lines 104, 109 and 112: Please, refer to Fig. 2.

Lines 106-108: Please, consider if this would suit better to the discussion part.

Figure 1

Line 117: Please, add “Acinetobacter phage” before PMK34.

Line 118: There is one extra bracket in the end of the sentence.

Figure 2:

Please consider to give a bit more informative title for the figure than “Multiple protein sequence alignment”. What proteins have been aligned? Why, what were the results? Not too long though.

Upper column of the figure contains bars “T7 phage tail fiber” and “pectate lyase_3 family”. Please describe these in the figure caption and add word “domain” in the end of both of these to make it clearer. Also add bar for the C-terminal domain. Since there is no T7 phage tail fiber sequence included in the figure, confusing.

Lines 131-132: Please change “T7 phage tail fiber domain” with “T7 phage tail fiber –like domain” or other appropriate since, as far as I understood, you don’t have the T7 phage tail fiber sequence in the Figure 2 analysis.

Line 133: Please add the explanations of the meaning of the different colors in the figure.

Results continues:

Line 138: Please, add the reference to Fig. 3.

Lines 139-140: Please, add a reference to the literature for “typical topology for depolymerases”.

Line 141: Inside the brackets, please remove the space in between the number and %, please, add the coverage information, what is 5js4.1? Please, add reference to the Fig. 3.

Line 145: Sentence “For reasons of simplicity, we coin DpoMK34 here as a tailspike.” This is rather confusing sentence, since you have usually used term DpoMK34, or depolymerase for this enzyme. Only in the figure 2 you have used term PMK34 tailspike. Please, be consistent and clear with the terminology.

Line 148: You use here symbol for ΦAB6 but not in the figure 2 (phiAB6). Which ever you choose, please be consistent with it.

Line 148: Please add reference to the Fig. 2 and reference to the literature for the sentence “Many structural studies of tail fibers and tailspikes are often limited to the N-terminal or C-terminal fragments due to the difficulty to obtain crystals…”.

Line 148: Please add reference to AlphaFold-Multimer.

Lines 154, 160 and 162: Please add references to the Fig. 4 (and to other suitable places, which I might have not noticed).

Figures 3 and 4:

Please consider combining these two figures since they both are about the AlphaFold-Multimer modelling and then name the different parts as 3a, 3b, 3c. I did not notice any references to the figure 3b (Predicted LDDT per position) in the text. Please correct, or remove this figure. Please add units (if applicable) to the x and y-axis of 3b figure (Predicted LDDT per position).

Line 164: Should you add reference to the AlphaFold-Multimer since you also have for Mol*3D here?

Line 170: Please open the abbreviation LDDT in the figure title and in the main text (line 501).

Figure 3: Please add explanations for the different colors used in the figure. To make it even clearer the names of the domains could be in the upper column of the figure and not just A, B and C, there is enough space for that.

Main text continues:

Line 179: Please, add the accession number for the genome of the phage PMK34.

Lines 180-182: Please remove spaces between figures and %.

Lines 178-184: In the methods you say you chose for BLASTp analysis hits with 100% query coverage and at least 94% identity (4.2), but it appears, this does not refer to the choice of the phage genomes here. Since, in the BLAST results there is not any other phage with 100% query coverage (except phage PMK34 itself). Why did you not choose for example Acinetobacter phage vB_AbaA_fBenAci001 (query coverage 87% and identity 94.55%) for this analysis? Since similarity and identity does not mean the same thing, talking about identities in the methods and similarities in the results is confusing to the reader. So please, describe in the methods in detail, how were these phages chosen, and maybe also shortly in the results text as well. And, please add the query coverages of these sequences compared to the phage PMK34 genome in addition to the similarities. To make it even clearer, I would recommend to have a table of the closely related phages including: the phage names, hosts, accession numbers, GC contents, genome sizes, query coverages compared to phage PMK34 and identities compared to the phage PMK34. Then you can only refer it in the text and make the text clearer. Also consider adding a phylogenetic tree of the phage relationships.

Many of these comments apply also to the choice of sequences in the Fig. 2 and Lines 191-196. Please make clearer to the reader how did you choose these sequences and please consider adding a table or a phylogenetic tree, if it clarifies these issues.  

Line 188: Please add a literature reference.

Line 199: Please add a reference to Fig. 5A.

Lines 202, 206 and 208: Please add literature references.

Line 202: When you mention Escherichia coli for the first time, you cannot use the abbreviation E. coli.

Figure 5:

Please add the explanations for the abbreviations of S, N, A, R, G and T in the figure 5A and/or in the text (Lines 197-199).

Figure 5A: text in the bottom right corner cannot be seen.

Figure 5A: Why were amino acids 651-699 chosen for the analysis?

Figure 5A: The term “consensus sequence S/A-S/A/R-N/G-A/T” would be nice in the figure caption also.

Figure 5B: names of the cyan coloured residues cannot be seen properly. Please add the names also to the Bottom view with font big enough. You don’t use the abbreviations GLY-683, ALA-681 etc. in the main text, why? Ideally these abbreviations should also be opened in the Figure caption.

Line 210: Please add a name/title for the Figure 5.

Line 210: Please open 4 aa: “four amino acids (aa)”.

Line 210: Please remove an extra space before word “C-terminal”.

Line 212: Please remove an extra space before the word “and”.

Main text:

Line 216: Have you tried to express and purify the enzyme with N-terminal His-tag or without a His-tag? This data would be interesting.

Line 225: How were the bacterial cells counted, could not find this in the methods.

Line 228: Please include that these are A. baumannii strains.

Lines 227-229: I would transfer the sentence: “No turbidity reduction….included as controls” to Line 223 before the sentences starting “Equal agar blocks…”. The results for these controls should be added in the figure 6B or then mentioned in the text that “data not shown”.

Figure 6:

Figure 6A: Please add the size of the depolymerase, 76 kDa, on the right hand side of the figure next to the band. What is the band with MW 29 kDa?

Figure 6B and 6C please refer to the Main issues: “In the beginning the authors talk about putative depolymerase. But, quite soon, they start to use term depolymerase without questioning it. To me as a reviewer, it is still a bit vague if this enzyme is a depolymerase or a lysin or what? My doubts started with the Figure 6. In figure 6B the putative depolymerase causes clearance of the host bacteria on an agar plate but not in the control sample. Interestingly, the cell counts are similar in the control sample and in the two different depolymerase concentration samples. How is this possible? Is the reasoning behind that the depolymerase cleaves the capsule and leaves the bacteria alive. But why don’t these bacteria cause turbidity on the plate anymore if they are still alive? Or dead, since even dead cells would cause turbidity. Unfortunately, I don’t understand this, could you please clarify the reasoning behind this in the discussion so that the reader does not have to guess how did you ended up to your conclusions. In addition, please describe in the methods how you determined the cell counts, I could not find this. Also, please consider adding light microscopy images as in Fig 1. for these three different samples presented in 6C, maybe this would further clarify this issue.”

Figure 6B: Please consider adding data for the controls RUH134, Greek46 and Greek 47 also (or maybe as supplemental material?).

Main text continues:

Line 247: What do you mean by semi-quantitative here?

Line 249: Please add a space in between 4 and °C. You use different °C symbol in different places. Please, choose one and stick to that.

Line 254: Please remove extra space between 10 and %

Figure 7:

Please add MHFC abbreviation in the y-axis title.

Lines 268-269: Is this the explanation also for the temperatures 4 °C, 40 °C and 50 °C missing the standard deviations? Please add this information in the figure caption, if so.

Main text continues:

Line 271: Please remove the dot in the end of the title (applies to other titles as well, at least lines: 88, 287, 302, 569).

Line 272: No space between 95 and %.

Line 272: Please add word “phage” before word “particles”.

Line 276: No space between 16 and %.

Line 276: Please add word “phage” before “particles”.

Lines: 276-279: These two last sentences might be better in the discussion. Please, consider this.

Figure 8:

Line 285: Please correct: “Student t-test” with “Student’s t-test”. Also in lines: 323 and 616.

Main text:

Lines 291-294: Please add references.

Line 294: Please open MIC if not opened earlier.  

Lines 309 and 313: Please add references to the Fig 9.

Figure 9:

Could you fit in explanations for the two different columns normal serum and heat inactivated. Maybe below the figure, but above the percentages?

Consider also using common term heat-inactivated serum (HIS) instead of heat deactivated serum and complement inactivated serum. Although, all can be understood well.

Please add if the % values are (v/v) or what?

Main text:

Line 331: Please add “data not shown” if this is the case.

Discussion

Please add more support and evidence in a clear way that DpoMK34 really is a depolymerase and not other type of enzyme.

Please add more references to the data of this study in the discussion, meaning references to the figures!

Line 342: Please add “In this study” before “We have identified..”

Line 345: How about reference Guo et al. 2017 for E. coli? (Identification and Characterization of Dpo42, a Novel Depolymerase Derived from the Escherichia coli Phage vB_EcoM_ECOO78)

Lines 348-352: This sentence is a bit too long and difficult to understand. Please clarify.

Line 357: Please add “a” before “result”.

Line 359: Reference to the relevant figure.

Lines 364-365: Where is this data? Reference?

Lines 378, 381, 382: References to the relevant figures.

Line 380: Please add “bacteriophage” before “PMK34”.

Page 12 of 20: Figure, literature and inner references to relevant paragraphs. No space between number and %.

Line 445: Please add reference to the relevant figure or “data not shown”.

Line 459: Please replace “its” with “their”

Lines 463-468: Please add in the conclusions also something about your findings in this study.

Materials and Methods:

Lines 474-475: Please use as outer brackets [ and ].

Line 480: Please add “phage” before “PMK34”.

Lines 482-483: “resulting in the …around the plaques” This was already in the results section and does not belong to the Materials and Methods.

Lines 484-485: Please add (v/v) or (w/v) wen needed.

Lines 487-488: Wasn’t the genomic sequence already published in your previous article (23)?

Lines 491-493: This belongs to the introduction not to the materials and methods.

Line 497: Please replace “size” with “weight”.

Line 501: Open abbreviations NVIDIA, GPU and LDDT.

Line 506: Please replace “cover” with “coverage”.

Line 515: Phusion is written with capital letter.

Line 517: Please add reference for the expression vector.

Line 529: Please open K MWCO abbreviation.

Line 531: Please add (w/v) after the percentage. Please open the abbreviation SDS-PAGE.

Line 537: Correct use of brackets! [()]

Paragraph 4.4: There is a lot of repetition, please summarize this paragraph.

Line 562: Please open abbreviation MOI (when you use it for the first time).

Line 593: correct use of brackets!

Reviewer 2 Report

Overall, the work is good. The experimental design is good and properly organized also.

Comments:

  1. What do you mean by the statement, “The literature tends to be somewhat ambiguous about the naming of tail fibers and tailspikes” in L. 141-142? If it should be there, restate it more clearly.
  2. Please italicize “Acinetobacter” in all its appearances, such as “Acinetobacter phage”.
  3. Please insert the reference for “Yehl and co-workers” in L. 206-208, and better to restate as “Yehl et al. [reference]”.
  4. Insert a comma after “After 16h at 35 ºC” in L. 221.
  5. Could you consider including at least one of the results for the controls (L. 227-229) in Fig. 6B?
  6. The argument that decapsulating A. baumannii MK34 cells with DpoMK34 does not sensitize them to the action of imipenem, amikacin and colistin better be illustrated by experimental data or figure

Round 2

Reviewer 1 Report

In the original article “The specific capsule depolymerase of phage PMK34 sensitizes Acinetobacter baumannii to serum killing” the authors have identified and analyzed a putative phage depolymerase encoding orf45, cloned, produced and purified the gene product and characterized the functions of this putative phage depolymerase.  

Comments:

Thank you for making such good progress with the manuscript!

Major issues:

Thank you for the clarifications! Please consider that the results and conclusions made in the article should be clear to the readers without special screen settings or printers. I would suggest you to describe your visual findings (shortly) in the results and discussion as you have described me here:

“The spots caused by DpoMK34 in figure 6B are not fully cleared (as would be the case for example when dropping a phage lysin, lysing and thus killing the cells) but only shows reduced turbidity. Surviving cells are easily visible in the spot with the naked eye. The loss in turbidity is observed for all depolymerase and is because all cells have lost their capsule. Capsule loss does not cause killing of the cells, only a less turbid appearance. The turbidity of these spots is similar to the turbidity of the growing halo’s surrounding a plaque as shown in figure 1A.”

Minor issues:

Line 29: To my understanding, phage names are written fully either in italics (taxonomic units) or normally (isolates). These two styles are not mixed in a one phage name. I am aware, the other reviewer asked to correct “Acinetobacter” in italics, but I dare to disagree on his/her opinion. So, please correct back to “Acinetobacter phage vB_AbaP_PMK34”.

Please correct this issue in each position where the full phage name appears.

Line 48: Please remove space between 10 and %.

Line 49: Please add reference, if it is [1], then please add it.

Line 54: Please add reference.

Line 114: Please add reference to Fig. 2.

Line 123: Could you very briefly tell (in parenthesis) what Maneval’s staining stains, what is it specific for?

Figure 2: The upper bars are more informative than previously, good! Just please change the font color to black, so that the text can be read from printed version.

Line 128: Please add “s” after “tailspike”

Lines 184-185: Please add the meaning of colours in the figure caption f.ex. “domains well (blue).” and “domains is high (red),”

Line 188: Please remove the space between number and %

Lines 194-197: Reference missing

Line 197: orf45 in italics

Line 222: Please add the pair for parentheses.

Line 244: Please add “s” to word “tailpike”

Figure 7: Please add the previous corrections to the figure caption.

Lines 304 and 305: Please replace the outermost parentheses with [].

Line 363: Please write “Autographiviridae” in italics.

Lines 374-376: Please, discuss more in detail how does your data give support to the assumption that DpoMK34 is a depolymerase and not f.ex. a lysin?

Lines 616, 617, 618: Please remove the space between number and %.

Author Response

Thank you for making such good progress with the manuscript!

We thank the reviewer for the appreciating words.

Major issues:

Thank you for the clarifications! Please consider that the results and conclusions made in the article should be clear to the readers without special screen settings or printers. I would suggest you describe your visual findings (shortly) in the results and discussion as you have described me here:

“The spots caused by DpoMK34 in figure 6B are not fully cleared (as would be the case for example when dropping a phage lysin, lysing and thus killing the cells) but only shows reduced turbidity. Surviving cells are easily visible in the spot with the naked eye. The loss in turbidity is observed for all depolymerase and is because all cells have lost their capsule. Capsule loss does not cause killing of the cells, only a less turbid appearance. The turbidity of these spots is similar to the turbidity of the growing halo’s surrounding a plaque as shown in figure 1A.”

The description of the observation has now been added to  the result section of the revised version to stress that these zones have a reduced turbidity but are not clear ones.

Line 233 - 237:

“The spots are not fully cleared (as would be the case for example when dropping a phage lysin, lysing and thus killing the cells) but shows reduced turbidity (similar to the reduced turbidity of halos growing around phage plaques). This loss in turbidity is a hallmark feature for all depolymerase linked to capsule loss.”

Minor issues:

Line 29: To my understanding, phage names are written fully either in italics (taxonomic units) or normally (isolates). These two styles are not mixed in a one phage name. I am aware, the other reviewer asked to correct “Acinetobacter” in italics, but I dare to disagree on his/her opinion. So, please correct back to “Acinetobacter phage vB_AbaP_PMK34”.

Please correct this issue in each position where the full phage name appears.

Corrections performed. 

Line 48: Please remove space between 10 and %.

Space removed. Apart from the unnecessary spaces listed here, spaces have been also removed in other cases where a space was still present.

Line 49: Please add reference, if it is [1], then please add it.

Reference added

Line 54: Please add reference.

Reference added

Line 114: Please add reference to Fig. 2.

A reference to Figure 2 has been added

Line 123: Could you very briefly tell (in parenthesis) what Maneval’s staining stains, what is it specific for?

Brief description added

Line 123-124:

“Maneval’s staining (specific capsular negative staining)”

Figure 2: The upper bars are more informative than previously, good! Just please change the font color to black, so that the text can be read from printed version.

The font colour was changed to black whereas  the box’s background was changed to improve contrast.

Line 128: Please add “s” after “tailspike”

Correction performed.

Lines 184-185: Please add the meaning of colours in the figure caption f.ex. “domains well (blue).” and “domains is high (red),”

Colour indications have been added.

Line 188: Please remove the space between number and %

Space removed. Apart from the unnecessary spaces listed here, spaces have been also removed in other cases where a space was still present.

Lines 194-197: Reference missing

The supporting example, Acinetobacter phage SH-Ab 15599, is only deposited in NCBI. Therefore, we have added the accession number.

Line 197: orf45 in italics

Correction performed.

Line 222: Please add the pair for parentheses.

Correction performed.

Line 244: Please add “s” to word “tailpike”

Tailspike here refers to the one of Acinetobacter phage B3 (singular)

Figure 7: Please add the previous corrections to the figure caption.

Corrections added

Lines 304 and 305: Please replace the outermost parentheses with [].

Correction performed.

Line 363: Please write “Autographiviridae” in italics.

Correction performed.

Lines 374-376: Please, discuss more in detail how does your data give support to the assumption that DpoMK34 is a depolymerase and not f.ex. a lysin?

Confirmation of both identity and function of DpoMK34 has now been discussed in the revised version.

Line 376 – 381:

“In the current study, we followed a sequential in silico and experimental analysis to confirm both identity and activity of DpoMK34 as a depolymerase. BlastP analysis revealed putative depolymerases with query coverage up to 100% and similarities > 95%, but with a hypervariable 4 aa hotspot in the most ultimate loop. Structure modelling predicted an elongated protein structure with a typical right-handed β-helical topology (Fig. 2), suitable for capsule puncturing. Moreover, this shape allows better scanning and recognition of complementary receptors expressed on the host surface [43]. The initial in silico annotation of DpoMK34 as a putative depolymerase is confirmed by observing zones of reduced turbidity (not clear zones) upon spotting purified DpoMK34 on soft agar seeded with A. baumannii MK34 (Fig. 6). The resulting zones have a reduced turbidity similar to the turbidity of halos surrounding Acinetobacter phage PMK34 plaques (Fig. 1B). Bacterial cell counting in the spots showed the absence of bacterial killing as would happen when the cells are lysed by a lysin.”

Lines 616, 617, 618: Please remove the space between number and %.

Spaces removed. Apart from the unnecessary spaces listed here, spaces have been also removed in other cases where a space was still present.